# Memory-Limited Partially Observable Stochastic Control and Its Mean-Field Control Approach

**DOI:** 10.3390/e24111599

**Published:** 2022-11-03

**Authors:** Takehiro Tottori, Tetsuya J. Kobayashi

**Affiliations:** 1Department of Mathematical Informatics, Graduate School of Information Science and Technology, The University of Tokyo, Tokyo 113-8654, Japan; 2Institute of Industrial Science, The University of Tokyo, Tokyo 153-8505, Japan; 3Department of Electrical Engineering and Information Systems, Graduate School of Engineering, The University of Tokyo, Tokyo 113-8654, Japan; 4Universal Biology Institute, The University of Tokyo, Tokyo 113-8654, Japan

**Keywords:** decision-making, optimal control, stochastic control, incomplete information, memory limitation, mean-field control

## Abstract

Control problems with incomplete information and memory limitation appear in many practical situations. Although partially observable stochastic control (POSC) is a conventional theoretical framework that considers the optimal control problem with incomplete information, it cannot consider memory limitation. Furthermore, POSC cannot be solved in practice except in special cases. In order to address these issues, we propose an alternative theoretical framework, memory-limited POSC (ML-POSC). ML-POSC directly considers memory limitation as well as incomplete information, and it can be solved in practice by employing the technique of mean-field control theory. ML-POSC can generalize the linear-quadratic-Gaussian (LQG) problem to include memory limitation. Because estimation and control are not clearly separated in the LQG problem with memory limitation, the Riccati equation is modified to the partially observable Riccati equation, which improves estimation as well as control. Furthermore, we demonstrate the effectiveness of ML-POSC for a non-LQG problem by comparing it with the local LQG approximation.

## 1. Introduction

Control problems of systems with incomplete information and memory limitation appear in many practical situations. These constraints become especially predominant when designing the control of small devices [1,2], and are important for understanding the control mechanisms of biological systems [3,4,5,6,7,8] because their sensors are extremely noisy and their controllers can only have severely limited memories.

Partially observable stochastic control (POSC) is a conventional theoretical framework that considers the optimal control problem with one of these constraints, namely, the incomplete information of the system state (Figure 1b) [9]. Because the POSC controller cannot completely observe the state of the system, it determines the control based on the noisy observation history of the state. POSC can be solved in principle [10,11,12] by converting it to a completely observable stochastic control (COSC) of the posterior probability of the state, as the posterior probability represents the sufficient statistics of the observation history. The posterior probability and the optimal control are obtained by solving the Zakai equation and the Bellman equation, respectively.

However, POSC has three practical problems with respect to the implementation of the controller which originate from the ignorance of the other constraint, namely, the memory limitation of the controller [1,2]. First, a controller designed by POSC should ideally have an infinite-dimensional memory to store and compute the posterior probability from the observation history. Second, the memory of the controller cannot have intrinsic stochasticity other than the observation noise to accurately compute the posterior probability via the Zakai equation. Third, POSC does not consider the cost originating from the memory update, which can be regarded as a cost of estimation. In light of the dualistic roles played by estimation and control, considering only control cost by ignoring estimation cost is asymmetric. As a result, POSC is not practical for control problems where the memory size, noise, and cost are non-negligible. Therefore, we need an alternative theoretical framework considering memory limitation to circumvent these three problems.

Furthermore, POSC has another crucial problem in obtaining the optimal state control by solving the Bellman equation [3,4]. Because the posterior probability of the state is infinite-dimensional, POSC corresponds to an infinite-dimensional COSC. In the infinite-dimensional COSC, the Bellman equation becomes a functional differential equation, which needs to be solved in order to obtain the optimal state control. However, solving a functional differential equation is generally intractable, even numerically.

In this work, we propose an alternative theoretical framework to the conventional POSC which can address the above-mentioned two issues. We call it memory-limited POSC (ML-POSC), in which memory limitation as well as incomplete information are directly accounted (Figure 1c). The conventional POSC derives the Zakai equation without considering memory limitations. Then, the optimal state control is supposed to be derived by solving the Bellman equation, even though we do not have any practical way to do this. In contrast, ML-POSC first postulates the finite-dimensional and stochastic memory dynamics explicitly by taking the memory limitation into account and then jointly optimizes the memory dynamics and state control by considering the memory and control costs. As a result, unlike the conventional POSC, ML-POSC finds both the optimal state control and the optimal memory dynamics with given memory limitations. Furthermore, we show that the Bellman equation of ML-POSC can be reduced to the Hamilton–Jacobi–Bellman (HJB) equation by employing a trick from the mean-field control theory [13,14,15]. While the Bellman equation is a functional differential equation, the HJB equation is a partial differential equation. As a result, ML-POSC can be solved, at least numerically.

The idea behind ML-POSC is closely related to that of the finite-state controller [16,17,18,19,20,21,22]. Finite-state controllers have been studied using the partially observable Markov decision process (POMDP), that is, the discrete time and state POSC. The finite-dimensional memory of ML-POSC can be regarded as an extension of the finite-state controller of POMDP to the continuous time and state setting. Nonetheless, the algorithms of the finite-state controller cannot be directly extended to the continuous setting, as they strongly depend on the discreteness. Although Fox and Tishby extended the finite-state controller to the continuous setting, their algorithm is restricted to the special case [1,2]. ML-POSC resolves this problem by employing the technique of the mean-field control theory.

In the linear-quadratic-Gaussian (LQG) problem of the conventional POSC, the Zakai equation and the Bellman equation are reduced to the Kalman filter and the Riccati equation, respectively [9,23]. Because the infinite-dimensional Zakai equation is reduced to the finite-dimensional Kalman filter, the LQG problem of the conventional POSC can be discussed in terms of ML-POSC. We show that the Kalman filter corresponds to the optimal memory dynamics of ML-POSC. Moreover, ML-POSC can generalize the LQG problem to include memory limitations such as the memory noise and cost. Because estimation and control are not clearly separated in the LQG problem with memory limitation, the Riccati equation for control is modified to include estimation, which in this paper is called the partially observable Riccati equation. We demonstrate that the partially observable Riccati equation is superior to the conventional Riccati equation as concerns the LQG problem with memory limitation.

Then, we investigate the potential effectiveness of ML-POSC for a non-LQG problem by comparing it with the local LQG approximation of the conventional POSC [3,4]. In the local LQG approximation, the Zakai equation and the Bellman equation are locally approximated by the Kalman filter and the Riccati equation, respectively. Because the Bellman equation (a functional differential equation) is reduced to the Riccati equation (an ordinary differential equation), the local LQG approximation can be solved numerically. However, the performance of the local LQG approximation may be poor in a highly non-LQG problem, as the local LQG approximation ignores non-LQG information. In contrast, ML-POSC reduces the Bellman equation to the HJB equation while maintaining non-LQG information. We demonstrate that ML-POSC can provide a better result than the local LQG approximation in a non-LQG problem.

This paper is organized as follows: In Section 2, we briefly review the conventional POSC. In Section 3, we formulate ML-POSC. In Section 4, we propose the mean-field control approach to ML-POSC. In Section 5, we investigate the LQG problem of the conventional POSC based on ML-POSC. In Section 6, we generalize the LQG problem to include memory limitation. In Section 7, we show numerical experiments involving a LQG problem with memory limitation and a non-LQG problem. Finally, in Section 8, we discuss our work.

## 2. Review of Partially Observable Stochastic Control

In this section, we briefly review the conventional POSC [11,15].

### 2.1. Problem Formulation

In this subsection, we formulate the conventional POSC [11,15]. The state $x_{t}\in R^{d_{x}}$ and the observation $y_{t}\in R^{d_{y}}$ at time t∈[0,T] evolve by the following stochastic differential equations (SDEs): (1)$$\begin{array}{cc} dx_{t} & =b\left(t,x_{t},u_{t}\right)dt+\sigma \left(t,x_{t},u_{t}\right)d\omega_{t}, \end{array}$$(2)$$\begin{array}{cc} dy_{t} & =h\left(t,x_{t}\right)dt+\gamma \left(t\right)d\nu_{t}, \end{array}$$
where $x_{0}$ and $y_{0}$ obey $p_{0}\left(x_{0}\right)$ and $p_{0}\left(y_{0}\right)$, respectively, $\omega_{t}\in R^{d_{\omega }}$ and $\nu_{t}\in R^{d_{\nu }}$ are independent standard Wiener processes, and $u_{t}\in R^{d_{u}}$ is the control. Here, $\gamma \left(t\right)\gamma^{⊤}\left(t\right)$ is assumed to be invertible. In POSC, because the controller cannot completely observe the state $x_{t}$, the control $u_{t}$ is determined based on the observation history $y_{0:t}:=\left\{y_{\tau }|\tau \in \left[0,t\right]\right\}$, as follows:(3)$$\begin{array}{c} u_{t}=u\left(t,y_{0:t}\right). \end{array}$$

The objective function of POSC is provided by the following expected cumulative cost function:(4)$$\begin{array}{c} J\left[u\right]:=E_{p(x_{0:T},y_{0:T};u)}\left[\int_{0}^{T}f\left(t,x_{t},u_{t}\right)dt+g\left(x_{T}\right)\right], \end{array}$$
where *f* is the cost function, *g* is the terminal cost function, $p(x_{0:T},y_{0:T};u)$ is the probability of $x_{0:T}$ and $y_{0:T}$ given *u* as a parameter, and $E_{p}\left[\cdot \right]$ is the expectation with respect to probability *p*. Throughout this paper, the time horizon *T* is assumed to be finite.

POSC is the problem of finding the optimal control function $u^{*}$ that minimizes the objective function J[u] as follows:(5)$$\begin{array}{c} u^{*}:=\underset{u}{\mathrm{argmin}}J\left[u\right]. \end{array}$$

### 2.2. Derivation of Optimal Control Function

In this subsection, we briefly review the derivation of the optimal control function of the conventional POSC [11,15]. We first define the unnormalized posterior probability density function $q_{t}\left(x\right):=p\left(x_{t}=x,y_{0:t}\right)$. We omit $y_{0:t}$ for notational simplicity. Here, $q_{t}\left(x\right)$ obeys the following Zakai equation:(6)$$\begin{array}{c} dq_{t}\left(x\right)=L^{†}q_{t}\left(x\right)dt+q_{t}\left(x\right)h^{⊤}\left(t,x\right){\left(\gamma \left(t\right)\gamma^{⊤}\left(t\right)\right)}^{-1}dy_{t}, \end{array}$$
where $q_{0}\left(x\right)=p_{0}\left(x\right)p_{0}\left(y\right)$ and $L^{†}$ is the forward diffusion operator, which is defined by
(7)$$\begin{array}{cc} L^{†}q\left(x\right) & :=-\sum_{i=1}^{d_{x}}\frac{\partial (b_{i}\left(t,x,u\right)q\left(x\right))}{\partial x_{i}}+\frac{1}{2}\sum_{i,j=1}^{d_{x}}\frac{\partial^{2}\left(D_{ij}\left(t,x,u\right)q\left(x\right)\right)}{\partial x_{i}\partial x_{j}}, \end{array}$$
where $D\left(t,x,u\right):=\sigma \left(t,x,u\right)\sigma^{⊤}\left(t,x,u\right)$. Then, the objective function (Equation 4) can be calculated as follows:(8)$$\begin{array}{c} J\left[u\right]=E_{p(q_{0:T};u)}\left[\int_{0}^{T}\bar{f}\left(t,q_{t},u_{t}\right)dt+\bar{g}\left(q_{T}\right)\right], \end{array}$$
where $\bar{f}\left(t,q,u\right):=E_{q(x)}\left[f(t,x,u)\right]$ and $\bar{g}\left(q\right):=E_{q(x)}\left[g(x)\right]$. From (Equation 6) and (Equation 8), POSC is converted into a COSC of $q_{t}$. As a result, POSC can be approached in the similar way as COSC, and the optimal control function is provided by the following proposition.

**Proposition 1** ([11,15])**.**
*The optimal control function of POSC is provided by*
(9)$$\begin{array}{c} u^{*}\left(t,q\right)=\underset{u}{\mathrm{argmin}}E_{q(x)}\left[H\left(t,x,u,\frac{\delta V(t,q)}{\delta q}\left(x\right)\right)\right], \end{array}$$
*where H is the Hamiltonian, which is defined by*
(10)$$\begin{array}{c} H\left(t,x,u,\frac{\delta V(t,q)}{\delta q}\left(x\right)\right):=f\left(t,x,u\right)+L\frac{\delta V(t,q)}{\delta q}\left(x\right). \end{array}$$
*L is the backward diffusion operator, which is defined by*

(11)
$$\begin{array}{cc} Lq(x) & :=\sum_{i=1}^{d_{x}}b_{i}\left(t,x,u\right)\frac{\partial q(x)}{\partial x_{i}}+\frac{1}{2}\sum_{i,j=1}^{d_{x}}D_{ij}\left(t,x,u\right)\frac{\partial^{2}q\left(x\right)}{\partial x_{i}\partial x_{j}}. \end{array}$$


*We note that L is the conjugate of $L^{†}$; furthermore, V(t,q) is the value function, which is the solution of the following Bellman equation:*

(12)
$$\begin{array}{cc} \; & -\frac{\partial V(t,q)}{\partial t}=E_{q(x)}\left[H\left(t,x,u^{*},\frac{\delta V(t,q)}{\delta q}\left(x\right)\right)\right]\\ \; & +\frac{1}{2}E_{q\left(x\right)q\left(x^{\prime }\right)}\left[\frac{\delta }{\delta q}\frac{\delta V(t,q)}{\delta q}\left(x,x^{\prime }\right)h^{⊤}\left(t,x\right){\left(\gamma \left(t\right)\gamma^{⊤}\left(t\right)\right)}^{-1}h\left(t,x^{\prime }\right)\right], \end{array}$$

*where $V\left(T,q\right)=E_{q(x)}\left[g(x)\right]$.*


**Proof.** The proof is shown in [11,15]. □

The optimal control function $u^{*}\left(t,q\right)$ is obtained by solving the Bellman Equation (Equation 12). The controller determines the optimal control $u_{t}^{*}=u^{*}\left(t,q_{t}\right)$ based on the posterior probability $q_{t}$. The posterior probability $q_{t}$ is obtained by solving the Zakai Equation (Equation 6). As a result, POSC can be solved in principle.

However, POSC has three practical problems with respect to the memory of the controller. First, the controller should have an infinite-dimensional memory to store and compute the posterior probability $q_{t}$ from the observation history $y_{0:t}$. Second, the memory of the controller cannot have intrinsic stochasticity other than the observation $dy_{t}$ to accurately compute the posterior probability $q_{t}$ via the Zakai Equation (Equation 6). Third, POSC does not consider the cost originating from the memory update, which can be regarded as a cost of estimation. In light of the dualistic roles played by estimation and control, considering only control cost by ignoring estimation cost is asymmetric. As a result, POCS is not practical for control problems where the memory size, noise, and cost are non-negligible.

Furthermore, POSC has another crucial problem in obtaining the optimal control function $u^{*}\left(t,q\right)$ by solving the Bellman Equation (Equation 12). Because the posterior probability *q* is infinite-dimensional, the associated Bellman Equation (Equation 12) becomes a functional differential equation. However, solving a functional differential equation is generally intractable even numerically. As a result, POCS cannot be solved in practice.

## 3. Memory-Limited Partially Observable Stochastic Control

In order to address the above-mentioned problems, we propose an alternative theoretical framework to the conventional POSC called ML-POSC. In this section, we formulate ML-POSC.

### 3.1. Problem Formulation

In this subsection, we formulate ML-POSC. ML-POSC determines the control $u_{t}$ based on the finite-dimensional memory $z_{t}\in R^{d_{z}}$ as follows:(13)$$\begin{array}{c} u_{t}=u\left(t,z_{t}\right). \end{array}$$

The memory dimension $d_{z}$ is determined not by the optimization but by the prescribed memory limitation of the controller to be used. Comparing (Equation 3) and (Equation 13), the memory $z_{t}$ can be interpreted as the compression of the observation history $y_{0:t}$. While the conventional POSC compresses the observation history $y_{0:t}$ into the infinite-dimensional posterior probability $q_{t}$, ML-POSC compresses it into the finite-dimensional memory $z_{t}$.

ML-POSC formulates the memory dynamics with the following SDE:(14)$$\begin{array}{c} dz_{t}=c\left(t,z_{t},v_{t}\right)dt+\kappa \left(t,z_{t},v_{t}\right)dy_{t}+\eta \left(t,z_{t},v_{t}\right)d\xi_{t}, \end{array}$$
where $z_{0}$ obeys $p_{0}\left(z_{0}\right)$, $\xi_{t}\in R^{d_{\xi }}$ is the standard Wiener process, and $v_{t}=v\left(t,z_{t}\right)\in R^{d_{v}}$ is the control for the memory dynamics. This memory dynamics has three important properties: (i) because it depends on the observation $dy_{t}$, the memory $z_{t}$ can be interpreted as the compression of the observation history $y_{0:t}$; (ii) because it depends on the standard Wiener process $d\xi_{t}$, ML-POSC can consider the memory noise explicitly; (iii) because it depends on the control $v_{t}$, it can be optimized through the control $v_{t}$.

The objective function of ML-POSC is provided by the following expected cumulative cost function:(15)$$\begin{array}{c} J\left[u,v\right]:=E_{p(x_{0:T},y_{0:T},z_{0:T};u,v)}\left[\int_{0}^{T}f\left(t,x_{t},u_{t},v_{t}\right)dt+g\left(x_{T}\right)\right]. \end{array}$$

Because the cost function *f* depends on the memory control $v_{t}$ as well as the state control $u_{t}$, ML-POSC can consider the memory control cost (state estimation cost) as well as the state control cost explicitly.

ML-POSC optimizes the state control function *u* and the memory control function *v* based on the objective function J[u,v], as follows:(16)$$\begin{array}{c} u^{*},v^{*}:=\underset{u,v}{\mathrm{argmin}}J\left[u,v\right]. \end{array}$$

ML-POSC first postulates the finite-dimensional and stochastic memory dynamics explicitly, then jointly optimizes the state and memory control function by considering the state and memory control cost. As a result, unlike the conventional POSC, ML-POSC can consider memory limitation as well as incomplete information.

### 3.2. Problem Reformulation

Although the formulation of ML-POSC in the previous subsection clarifies its relationship with that of the conventional POSC, it is inconvenient for further mathematical investigations. In order to resolve this problem, we reformulate ML-POSC in this subsection. The formulation in this subsection is simpler and more general than that in the previous subsection.

We first define the extended state $s_{t}$ as follows:(17)$$\begin{array}{c} s_{t}:=\left(\begin{array}{c} x_{t}\\ z_{t} \end{array}\right)\in R^{d_{s}}, \end{array}$$
where $d_{s}=d_{x}+d_{z}$. The extended state $s_{t}$ evolves by the following SDE:(18)$$\begin{array}{c} ds_{t}=\tilde{b}\left(t,s_{t},{\tilde{u}}_{t}\right)dt+\tilde{\sigma }\left(t,s_{t},{\tilde{u}}_{t}\right)d{\tilde{\omega }}_{t}, \end{array}$$
where $s_{0}$ obeys $p_{0}\left(s_{0}\right)$, ${\tilde{\omega }}_{t}\in R^{d_{\tilde{\omega }}}$ is the standard Wiener process, and ${\tilde{u}}_{t}\in R^{d_{\tilde{u}}}$ is the control. ML-POSC determines the control ${\tilde{u}}_{t}\in R^{d_{\tilde{u}}}$ based solely on the memory $z_{t}$, as follows:(19)$$\begin{array}{c} {\tilde{u}}_{t}=\tilde{u}\left(t,z_{t}\right). \end{array}$$

The extended state SDE (Equation 18) includes the previous state, observation, and memory SDEs (Equation 1), (2) and (Equation 14) as a special case; they can be represented as follows:(20)$$ds_{t}=\left(\begin{array}{c} b(t,x_{t},u_{t})\\ c\left(t,z_{t},v_{t}\right)+\kappa \left(t,z_{t},v_{t}\right)h\left(t,x_{t}\right) \end{array}\right)dt\;+\left(\begin{array}{ccc} \sigma (t,x_{t},u_{t}) & O & O\\ O & \kappa \left(t,z_{t},v_{t}\right)\gamma \left(t\right) & \eta (t,z_{t},v_{t}) \end{array}\right)\left(\begin{array}{c} d\omega_{t}\\ d\nu_{t}\\ d\xi_{t} \end{array}\right),$$
where $p_{0}\left(s_{0}\right)=p_{0}\left(x_{0}\right)p_{0}\left(z_{0}\right)$.

The objective function of ML-POSC is provided by the following expected cumulative cost function:(21)$$\begin{array}{c} J\left[\tilde{u}\right]:=E_{p(s_{0:T};\tilde{u})}\left[\int_{0}^{T}\tilde{f}\left(t,s_{t},{\tilde{u}}_{t}\right)dt+\tilde{g}\left(s_{T}\right)\right], \end{array}$$
where $\tilde{f}$ is the cost function and $\tilde{g}$ is the terminal cost function. It is obvious that this objective function (Equation 21) is more general than the previous one (Equation 15).

ML-POSC is the problem of finding the optimal control function ${\tilde{u}}^{*}$ that minimizes the objective function $J[\tilde{u}]$ as follows:(22)$$\begin{array}{c} {\tilde{u}}^{*}:=\underset{\tilde{u}}{\mathrm{argmin}}J\left[\tilde{u}\right]. \end{array}$$

In the following section, we mainly consider the formulation in this subsection rather than that of the previous subsection, as it is simpler and more general. Moreover, we omit $\tilde{\cdot }$ for the notational simplicity.

## 4. Mean-Field Control Approach

If the control $u_{t}$ is determined based on the extended state $s_{t}$, i.e., $u_{t}=u\left(t,s_{t}\right)$, ML-POSC is the same as COSC of the extended state $s_{t}$, and can be solved by the conventional COSC approach [10]. However, because ML-POSC determines the control $u_{t}$ based solely on the memory $z_{t}$, i.e., $u_{t}=u\left(t,z_{t}\right)$, ML-POSC cannot be solved in a similar way as COSC. In order to solve ML-POSC, we propose the mean-field control approach in this section. Because the mean-field control approach is more general than the COSC approach, it can solve COSC and ML-POSC in a unified way.

### 4.1. Derivation of Optimal Control Function

In this subsection, we propose the mean-field control approach to ML-POSC. We first show that ML-POSC can be converted into a deterministic control of the probability density function, which is similar to the conventional POSC [11,15]. This approach is used in the mean-field control as well [13,14,24,25]. The extended state SDE (Equation 18) can be converted into the following Fokker–Planck (FP) equation:(23)$$\begin{array}{c} \frac{\partial p_{t}\left(s\right)}{\partial t}=L^{†}p_{t}\left(s\right), \end{array}$$
where the initial condition is provided by $p_{0}\left(s\right)$ and the forward diffusion operator $L^{†}$ is defined by (Equation 7). The objective function of ML-POSC (Equation 21) can be calculated as follows:(24)$$\begin{array}{c} J\left[u\right]=\int_{0}^{T}\bar{f}\left(t,p_{t},u_{t}\right)dt+\bar{g}\left(p_{T}\right), \end{array}$$
where $\bar{f}\left(t,p,u\right):=E_{p(s)}\left[f\left(t,s,u\right)\right]$ and $\bar{g}\left(p\right):=E_{p(s)}\left[g\left(s\right)\right]$. From (Equation 23) and (Equation 24), ML-POSC is converted into a deterministic control of $p_{t}$. As a result, ML-POSC can be approached in a similar way as the deterministic control, and the optimal control function is provided by the following lemma.

**Lemma 1.** 
*The optimal control function of ML-POSC is provided by*

(25)
$$\begin{array}{c} u^{*}\left(t,z\right)=\underset{u}{\mathrm{argmin}}E_{p_{t}\left(x|z\right)}\left[H\left(t,s,u,\frac{\delta V(t,p_{t})}{\delta p}\left(s\right)\right)\right], \end{array}$$

*where H is the Hamiltonian (Equation 10), $p_{t}\left(x|z\right)=p_{t}\left(s\right)/\int p_{t}\left(s\right)dx$ is the conditional probability density function of a state x given memory z, $p_{t}\left(s\right)$ is the solution of the FP Equation (Equation 23), and V(t,p) is the solution of the following Bellman equation:*

(26)
$$\begin{array}{c} -\frac{\partial V(t,p)}{\partial t}=E_{p(s)}\left[H\left(t,s,u^{*},\frac{\delta V(t,p)}{\delta p}\left(s\right)\right)\right], \end{array}$$

*where $V\left(T,p\right)=E_{p(s)}\left[g\left(s\right)\right]$.*


**Proof.** The proof is shown in Appendix A. □

The controller of ML-POSC determines the optimal control $u_{t}^{*}=u^{*}\left(t,z_{t}\right)$ based on the memory $z_{t}$, not the posterior probability $q_{t}$. Therefore, ML-POSC can consider memory limitation as well as incomplete information.

However, because the Bellman Equation (Equation 26) is a functional differential equation, it cannot be solved, even numerically, which is the same problem as the conventional POSC. We resolve this problem by employing the technique of the mean-field control theory [13,14] as follows.

**Theorem 1.** 
*The optimal control function of ML-POSC is provided by*

(27)
$$\begin{array}{c} u^{*}\left(t,z\right)=\underset{u}{\mathrm{argmin}}E_{p_{t}\left(x|z\right)}\left[H\left(t,s,u,w(t,s)\right)\right], \end{array}$$

*where H is the Hamiltonian (Equation 10), $p_{t}\left(x|z\right)=p_{t}\left(s\right)/\int p_{t}\left(s\right)dx$ is the conditional probability density function of a state x given memory z, $p_{t}\left(s\right)$ is the solution of the FP Equation (Equation 23), and w(t,s) is the solution of the following Hamilton–Jacobi–Bellman (HJB) equation:*

(28)
$$\begin{array}{c} -\frac{\partial w(t,s)}{\partial t}=H\left(t,s,u^{*},w\left(t,s\right)\right), \end{array}$$

*where w(T,s)=g(s).*


**Proof.** The proof is shown in Appendix B. □

While the Bellman Equation (Equation 26) is a functional differential equation, the HJB Equation (Equation 28) is a partial differential equation. As a result, unlike the conventional POSC, ML-POSC can be solved in practice.

We note that the mean-field control technique is applicable to the conventional POSC as well, and we obtain the HJB equation of the conventional POSC [15]. However, the HJB equation of the conventional POSC is not closed by a partial differential equation due to the last term of the Bellman Equation (Equation 12). As a result, the mean-field control technique is not effective with the conventional POSC except in a special case [15].

In the conventional POSC, the state estimation (memory control) and the state control are clearly separated. As a result, the state estimation and the state control are optimized by the Zakai Equation (Equation 6) and the Bellman Equation (Equation 12), respectively. In contrast, because ML-POSC considers memory limitation as well as incomplete information, the state estimation and the state control are not clearly separated. As a result, ML-POSC jointly optimizes the state estimation and the state control based on the FP Equation (Equation 23) and the HJB Equation (Equation 28).

### 4.2. Comparison with Completely Observable Stochastic Control

In this subsection, we show the similarities and differences between ML-POSC and COSC of the extended state. While ML-POSC determines the control $u_{t}$ based solely on the memory $z_{t}$, i.e., $u_{t}=u\left(t,z_{t}\right)$, COSC of the extended state determines the control $u_{t}$ based on the extended state $s_{t}$, i.e., $u_{t}=u\left(t,s_{t}\right)$. The optimal control function of COSC of the extended state is provided by the following proposition.

**Proposition 2** ([10])**.**
*The optimal control function of COSC of the extended state is provided by*
(29)$$\begin{array}{c} u^{*}\left(t,s\right)=\underset{u}{\mathrm{argmin}}H\left(t,s,u,w(t,s)\right), \end{array}$$
*where H is the Hamiltonian (Equation 10) and w(t,s) is the solution of the HJB Equation (Equation 28).*

**Proof.** The conventional proof is shown in [10]. We note that it can be proven in a similar way as ML-POSC, which is shown in Appendix C. □

Although the HJB Equation (Equation 28) is the same between ML-POSC and COSC, the optimal control function is different. While the optimal control function of COSC is provided by the minimization of the Hamiltonian (Equation 29), that of ML-POSC is provided by the minimization of the conditional expectation of the Hamiltonian (Equation 27). This is reasonable, as the controller of ML-POSC needs to estimate the state from the memory.

### 4.3. Numerical Algorithm

In this subsection, we briefly explain a numerical algorithm to obtain the optimal control function of ML-POSC (Equation 27). Because the optimal control function of COSC (Equation 29) depends only on the backward HJB Equation (Equation 28), it can be obtained by solving the HJB equation backwards from the terminal condition [10,26,27]. In contrast, because the optimal control function of ML-POSC (Equation 27) depends on the forward FP Equation (Equation 23) as well as the backward HJB Equation (Equation 28), it cannot be obtained in a similar way as COSC. Because the backward HJB equation depends on the forward FP equation through the optimal control function of ML-POSC, the HJB equation cannot be solved backwards from the terminal condition. As a result, ML-POSC needs to solve the system of HJB-FP equations.

The system of HJB-FP equations appears in the mean-field game and control [28,29,30], and many numerical algorithms have been developed [31,32,33]. Therefore, unlike the conventional POSC, ML-POSC can be solved in practice using these algorithms. Furthermore, unlike the mean-field game and control, the coupling of HJB-FP equations is limited to the optimal control function in ML-POSC. By exploiting this property, more efficient algorithms may be proposed for ML-POSC [34].

In this paper, we use the forward–backward sweep method (the fixed-point iteration method) to obtain the optimal control function of ML-POSC [33,34,35,36,37], which is one of the most basic algorithms for the system of HJB-FP equations. The forward–backward sweep method computes the forward FP Equation (Equation 23) and the backward HJB Equation (Equation 28) alternately. In the mean-field game and control, the convergence of the forward–backward sweep method is not guaranteed. In contrast, it is guaranteed in ML-POSC because the coupling of HJB-FP equations is limited to the optimal control function [34].

## 5. Linear-Quadratic-Gaussian Problem without Memory Limitation

In the LQG problem of the conventional POSC, the Zakai Equation (Equation 6) and the Bellman Equation (Equation 12) are reduced to the Kalman filter and the Riccati equation, respectively [9,23]. Because the infinite-dimensional Zakai equation is reduced to the finite-dimensional Kalman filter, the LQG problem of the conventional POSC can be discussed in terms of ML-POSC. In this section, we briefly review the LQG problem of the conventional POSC, then reproduce the Kalman filter and the Riccati equation from the viewpoint of ML-POSC. The LQG problem of the conventional POSC corresponds to the LQG problem without memory limitation, as it does not consider the memory noise and cost.

### 5.1. Review of Partially Observable Stochastic Control

In this subsection, we briefly review the LQG problem of the conventional POSC [9,23]. The state $x_{t}\in R^{d_{x}}$ and the observation $y_{t}\in R^{d_{y}}$ at time t∈[0,T] evolve by the following SDEs: (30)$$\begin{array}{cc} dx_{t} & =\left(A\left(t\right)x_{t}+B\left(t\right)u_{t}\right)dt+\sigma \left(t\right)d\omega_{t}, \end{array}$$(31)$$\begin{array}{cc} dy_{t} & =H\left(t\right)x_{t}dt+\gamma \left(t\right)d\nu_{t}, \end{array}$$
where $x_{0}$ obeys the Gaussian distribution $p_{0}\left(x_{0}\right)=N\left(x_{0}\left|\mu_{x,0},\Sigma_{xx,0}\right.\right)$, $y_{0}$ is an arbitrary real vector, $\omega_{t}\in R^{d_{\omega }}$ and $\nu_{t}\in R^{d_{\nu }}$ are independent standard Wiener processes, and $u_{t}=u\left(t,y_{0:t}\right)\in R^{d_{u}}$ is the control. Here, $\gamma \left(t\right)\gamma^{⊤}\left(t\right)$ is assumed to be invertible. The objective function is provided by the following expected cumulative cost function:(32)$$\begin{array}{c} J\left[u\right]:=E_{p(x_{0:T},y_{0:T};u)}\left[\int_{0}^{T}\left(x_{t}^{⊤}Q\left(t\right)x_{t}+u_{t}^{⊤}R\left(t\right)u_{t}\right)dt+x_{T}^{⊤}Px_{T}\right], \end{array}$$
where Q(t)⪰O, R(t)≻O, and P⪰O. The LQG problem of the conventional POSC is to find the optimal control function $u^{*}$ that minimizes the objective function J[u], as follows:(33)$$\begin{array}{c} u^{*}:=\underset{u}{\mathrm{argmin}}J\left[u\right]. \end{array}$$

In the LQG problem of the conventional POSC, the posterior probability is provided by the Gaussian distribution $p\left(x_{t}|y_{0:t}\right)=N\left(x_{t}|\overset{ˇ}{\mu }\left(t\right),\overset{ˇ}{\Sigma }\left(t\right)\right)$, and $u_{t}=u\left(t,y_{0:t}\right)$ is reduced to $u_{t}=u\left(t,{\overset{ˇ}{\mu }}_{t}\right)$ without loss of performance.

**Proposition 3** ([9,23])**.**
*In the LQG problem without memory limitation, the optimal control function of POSC (Equation 33) is provided by*
(34)$$\begin{array}{c} u^{*}\left(t,\overset{ˇ}{\mu }\right)=-R^{-1}B^{⊤}\Psi \overset{ˇ}{\mu }, \end{array}$$
*where $\overset{ˇ}{\mu }\left(t\right)$ and $\overset{ˇ}{\Sigma }\left(t\right)$ are the solutions of the following Kalman filter:*
(35)$$\begin{array}{cc} \; & d\overset{ˇ}{\mu }=\left(A-BR^{-1}B^{⊤}\Psi \right)\overset{ˇ}{\mu }dt+\overset{ˇ}{\Sigma }H^{⊤}{\left(\gamma \gamma^{⊤}\right)}^{-1}\left(dy_{t}-H\overset{ˇ}{\mu }dt\right), \end{array}$$
(36)$$\begin{array}{cc} \; & \frac{d\overset{ˇ}{\Sigma }}{dt}=\sigma \sigma^{⊤}+A\overset{ˇ}{\Sigma }+\overset{ˇ}{\Sigma }A^{⊤}-\overset{ˇ}{\Sigma }H^{⊤}{\left(\gamma \gamma^{⊤}\right)}^{-1}H\overset{ˇ}{\Sigma }, \end{array}$$
*and where $\overset{ˇ}{\mu }\left(0\right)=\mu_{x,0}$ and $\overset{ˇ}{\Sigma }\left(0\right)=\Sigma_{xx,0}$. Ψ(t) is the solution of the following Riccati equation:*
(37)$$\begin{array}{cc} -\frac{d\Psi }{dt} & =Q+A^{⊤}\Psi +\Psi A-\Psi BR^{-1}B^{⊤}\Psi , \end{array}$$
*where Ψ(T)=P.*

**Proof.** The proof is shown in [9,23]. □

In the LQG problem of the conventional POSC, the Zakai Equation (Equation 6) and the Bellman Equation (Equation 12) are reduced to the Kalman filter (Equation 35) and (Equation 36) and the Riccati Equation (Equation 37), respectively.

### 5.2. Memory-Limited Partially Observable Stochastic Control

Because the infinite-dimensional Zakai Equation (Equation 6) is reduced to the finite-dimensional Kalman filter (Equation 35) and (Equation 36), the LQG problem of the conventional POSC can be discussed in terms of ML-POSC. In this subsection, we reproduce the Kalman filter (Equation 35) and (Equation 36) and the Riccati Equation (Equation 37) from the viewpoint of ML-POSC.

ML-POSC defines the finite-dimensional memory $z_{t}\in R^{d_{z}}$. In the LQG problem of the conventional POSC, the memory dimension $d_{z}$ is the same as the state dimension $d_{x}$. The controller of ML-POSC determines the control $u_{t}$ based on the memory $z_{t}$, i.e., $u_{t}=u\left(t,z_{t}\right)$. The memory $z_{t}$ is assumed to evolve by the following SDE:(38)$$\begin{array}{c} dz_{t}=v_{t}dt+\kappa_{t}dy_{t}, \end{array}$$
where $z_{0}=\mu_{0,xx}$, while $v_{t}=v\left(t,z_{t}\right)\in R^{d_{z}}$ and $\kappa_{t}=\kappa \left(t,z_{t}\right)\in R^{d_{z}\times d_{y}}$ are the memory controls. We note that the LQG problem of the conventional POSC does not consider the memory noise. The objective function of ML-POSC is provided by the following expected cumulative cost function:(39)$$\begin{array}{c} J\left[u,v,\kappa \right]:=E_{p(x_{0:T},y_{0:T},z_{0:T};u,v,\kappa )}\left[\int_{0}^{T}\left(x_{t}^{⊤}Q\left(t\right)x_{t}+u_{t}^{⊤}R\left(t\right)u_{t}\right)dt+x_{T}^{⊤}Px_{T}\right]. \end{array}$$

We note that the LQG problem of the conventional POSC does not consider the memory control cost. ML-POSC optimizes *u*, *v*, and κ based on J[u,v,κ], as follows:(40)$$\begin{array}{c} u^{*},v^{*},\kappa^{*}:=\underset{u,v,\kappa }{\mathrm{argmin}}J\left[u,v,\kappa \right]. \end{array}$$

In the LQG problem of the conventional POSC, the probability of the extended state $s_{t}$ (Equation 17) is provided by the Gaussian distribution $p_{t}\left(s_{t}\right)=N\left(s_{t}|\mu \left(t\right),\Sigma \left(t\right)\right)$. The posterior probability of the state $x_{t}$ given the memory $z_{t}$ is provided by the Gaussian distribution $p_{t}\left(x_{t}|z_{t}\right)=N\left(x_{t}|\mu_{x|z}\left(t,z_{t}\right),\Sigma_{x|z}\left(t\right)\right)$, where $\mu_{x|z}\left(t,z_{t}\right)$ and $\Sigma_{x|z}\left(t\right)$ are provided as follows: (41)$$\begin{array}{cc} \mu_{x|z}\left(t,z_{t}\right) & =\mu_{x}\left(t\right)+\Sigma_{xz}\left(t\right)\Sigma_{zz}^{-1}\left(t\right)\left(z_{t}-\mu_{z}\left(t\right)\right), \end{array}$$(42)$$\begin{array}{cc} \Sigma_{x|z}\left(t\right) & =\Sigma_{xx}\left(t\right)-\Sigma_{xz}\left(t\right)\Sigma_{zz}^{-1}\left(t\right)\Sigma_{zx}\left(t\right). \end{array}$$

**Theorem 2.** 
*In the LQG problem without memory limitation, the optimal control functions of ML-POSC (Equation 40) are provided by*

(43)
$$\begin{array}{cc} u^{*}\left(t,z\right) & =-R^{-1}B^{⊤}\Psi z, \end{array}$$


(44)
$$\begin{array}{cc} v^{*}\left(t,z\right) & =\left(A-BR^{-1}B^{⊤}\Psi -\Sigma_{x|z}H^{⊤}{\left(\gamma \gamma^{⊤}\right)}^{-1}H\right)z, \end{array}$$


(45)
$$\begin{array}{cc} \kappa^{*}\left(t,z\right) & =\Sigma_{x|z}H^{⊤}{\left(\gamma \gamma^{⊤}\right)}^{-1}. \end{array}$$


*From $v^{*}\left(t,z\right)$ and $\kappa^{*}\left(t,z\right)$, $z_{t}$ and $\Sigma_{x|z}\left(t\right)$ obey the following equations:*

(46)
$$\begin{array}{cc} \; & dz_{t}=\left(A-BR^{-1}B^{⊤}\Psi \right)z_{t}dt+\Sigma_{x|z}H^{⊤}{\left(\gamma \gamma^{⊤}\right)}^{-1}\left(dy_{t}-Hz_{t}dt\right), \end{array}$$


(47)
$$\begin{array}{cc} \; & \frac{d\Sigma_{x|z}}{dt}=\sigma \sigma^{⊤}+A\Sigma_{x|z}+\Sigma_{x|z}A^{⊤}-\Sigma_{x|z}H^{⊤}{\left(\gamma \gamma^{⊤}\right)}^{-1}H\Sigma_{x|z}, \end{array}$$

*where $z_{0}=\mu_{x,0}$ and $\Sigma_{x|z}\left(0\right)=\Sigma_{xx,0}$. Furthermore, $\mu_{x|z}\left(t,z_{t}\right)=z_{t}$ holds in this problem. Ψ(t) is the solution of the Riccati Equation (Equation 37).*


**Proof.** The proof is shown in Appendix D. □

In the LQG problem of the conventional POSC, the optimal memory dynamics of ML-POSC (Equation 46) and (Equation 47) corresponds to the Kalman filter (Equation 35) and (Equation 36). Furthermore, ML-POSC reproduces the Riccati Equation (Equation 37).

## 6. Linear-Quadratic-Gaussian Problem with Memory Limitation

The LQG problem of the conventional POSC does not consider memory limitation because it does not consider the memory noise and cost. Furthermore, because the memory dimension is restricted to the state dimension, the memory dimension cannot be determined according to a given controller. ML-POSC can generalize the LQG problem to include the memory limitation. In this section, we discuss the LQG problem with memory limitation based on ML-POSC.

### 6.1. Problem Formulation

In this subsection, we formulate the LQG problem with memory limitation. The state and observation SDEs are the same as in the previous section, which are provided by (Equation 30) and (Equation 31), respectively. The controller of ML-POSC determines the control $u_{t}\in R^{d_{u}}$ based on the memory $z_{t}\in R^{d_{z}}$, i.e., $u_{t}=u\left(t,z_{t}\right)$. Unlike the LQG problem of the conventional POSC, the memory dimension $d_{z}$ is not necessarily the same as the state dimension $d_{x}$.

The memory $z_{t}$ is assumed to evolve according to the following SDE:(48)$$\begin{array}{cc} dz_{t} & =v_{t}dt+\kappa \left(t\right)dy_{t}+\eta \left(t\right)d\xi_{t}, \end{array}$$
where $z_{0}$ obeys the Gaussian distribution $p_{0}\left(z_{0}\right)=N\left(z_{0}\left|\mu_{z,0},\Sigma_{zz,0}\right.\right)$, $\xi_{t}\in R^{d_{\xi }}$ is the standard Wiener process, and $v_{t}=v\left(t,z_{t}\right)\in R^{d_{v}}$ is the control. Because the initial condition $z_{0}$ is stochastic and the memory SDE (Equation 48) includes the intrinsic stochasticity $d\xi_{t}$, the LQG problem of ML-POSC can consider the memory noise explicitly. We note that κ(t) is independent of the memory $z_{t}$. If κ(t) depends on the memory $z_{t}$, the memory SDE (Equation 48) becomes non-linear and non-Gaussian. As a result, the optimal control functions cannot be derived explicitly in this case. In order to keep the memory SDE (Equation 48) linear and Gaussian for obtaining the optimal control functions explicitly, we restrict κ(t) being independent of the memory $z_{t}$ in the LQG problem with memory limitation. The LQG problem without memory limitation is the special case in which the optimal control $\kappa_{t}^{*}=\kappa^{*}\left(t,z_{t}\right)$ in (Equation 45) does not depend on the memory $z_{t}$.

The objective function is provided by the following expected cumulative cost function:(49)$$\begin{array}{c} J\left[u,v\right]:=E_{p(x_{0:T},y_{0:T},z_{0:T};u,v)}\left[\int_{0}^{T}\left(x_{t}^{⊤}Q\left(t\right)x_{t}+u_{t}^{⊤}R\left(t\right)u_{t}+v_{t}^{⊤}M\left(t\right)v_{t}\right)dt+x_{T}^{⊤}Px_{T}\right], \end{array}$$
where Q(t)⪰O, R(t)≻O, M(t)≻O, and P⪰O. Because the cost function includes $v_{t}^{⊤}M\left(t\right)v_{t}$, the LQG problem of ML-POSC can consider the memory control cost explicitly. ML-POSC optimizes the state control function *u* and the memory control function *v* based on the objective function J[u,v], as follows:(50)$$\begin{array}{c} u^{*},v^{*}:=\underset{u,v}{\mathrm{argmin}}J\left[u,v\right]. \end{array}$$

For the sake of simplicity, we do not optimize κ(t), although this can be accomplished by considering unobservable stochastic control.

### 6.2. Problem Reformulation

Although the formulation of the LQG problem with memory limitation in the previous subsection clarifies its relationship with that of the LQG problem without memory limitation, it is inconvenient for further mathematical investigations. In order to resolve this problem, we reformulate the LQG problem with memory limitation based on the extended state $s_{t}$ (Equation 17). The formulation in this subsection is simpler and more general than that in the previous subsection.

In the LQG problem with memory limitation, the extended state SDE (Equation 18) is provided as follows:(51)$$\begin{array}{c} ds_{t}=\left(\tilde{A}\left(t\right)s_{t}+\tilde{B}\left(t\right){\tilde{u}}_{t}\right)dt+\tilde{\sigma }\left(t\right)d{\tilde{\omega }}_{t}, \end{array}$$
where $s_{0}$ obeys the Gaussian distribution $p_{0}\left(s_{0}\right):=N\left(s_{0}\left|\mu_{0},\Sigma_{0}\right.\right)$, ${\tilde{\omega }}_{t}\in R^{d_{\tilde{\omega }}}$ is the standard Wiener process, and ${\tilde{u}}_{t}=\tilde{u}\left(t,z_{t}\right)\in R^{d_{\tilde{u}}}$ is the control. The extended state SDE (Equation 51) includes the previous state, observation, and memory SDEs (Equation 30), (Equation 31) and (Equation 48) as a special case because they can be represented as follows:(52)$$ds_{t}\;=\left(\left(\begin{array}{cc} A & O\\ \kappa H & O \end{array}\right)s_{t}+\left(\begin{array}{cc} B & O\\ O & I \end{array}\right){\tilde{u}}_{t}\right)dt+\left(\begin{array}{ccc} \sigma & O & O\\ O & \kappa \gamma & \eta \end{array}\right)\left(\begin{array}{c} d\omega_{t}\\ d\nu_{t}\\ d\xi_{t} \end{array}\right),$$
where $p_{0}\left(s_{0}\right)=p_{0}\left(x_{0}\right)p_{0}\left(z_{0}\right)$.

The objective function (Equation 21) is provided by the following expected cumulative cost function:(53)$$\begin{array}{c} J\left[\tilde{u}\right]:=E_{p(s_{0:T};\tilde{u})}\left[\int_{0}^{T}\left(s_{t}^{⊤}\tilde{Q}\left(t\right)s_{t}+{\tilde{u}}_{t}^{⊤}\tilde{R}\left(t\right){\tilde{u}}_{t}\right)dt+s_{T}^{⊤}\tilde{P}s_{T}\right], \end{array}$$
where $\tilde{Q}\left(t\right)⪰O$, $\tilde{R}\left(t\right)≻O$, and $\tilde{P}⪰O$. This objective function (Equation 53) includes the previous objective function (Equation 49) as a special case because it can be represented as follows:(54)$$J[\tilde{u}]=E_{p(s_{0:T};\tilde{u})}\left[\int_{0}^{T}\left(s_{t}^{⊤}\left(\begin{array}{cc} Q & O\\ O & O \end{array}\right)s_{t}+{\tilde{u}}_{t}^{⊤}\left(\begin{array}{cc} R & O\\ O & M \end{array}\right){\tilde{u}}_{t}\right)dt+s_{T}^{⊤}\left(\begin{array}{cc} P & O\\ O & O \end{array}\right)s_{T}\right].$$

The objective of the LQG problem with memory limitation is to find the optimal control function ${\tilde{u}}^{*}$ that minimizes the objective function $J[\tilde{u}]$, as follows:(55)$$\begin{array}{c} {\tilde{u}}^{*}:=\underset{\tilde{u}}{\mathrm{argmin}}J\left[\tilde{u}\right]. \end{array}$$

In the following subsection, we mainly consider the formulation of this subsection rather than that of the previous subsection because it is simpler and more general. Moreover, we omit $\tilde{\cdot }$ for notational simplicity.

### 6.3. Derivation of Optimal Control Function

In this subsection, we derive the optimal control function of the LQG problem with memory limitation by applying Theorem 1. In the LQG problem with memory limitation, the probability of the extended state *s* at time *t* is provided by the Gaussian distribution $p_{t}\left(s\right)=N\left(s|\mu (t),\Sigma (t)\right)$. By defining the stochastic extended state $\hat{s}:=s-\mu$, $E_{p_{t}\left(x|z\right)}\left[s\right]$ is provided as follows:(56)$$\begin{array}{c} E_{p_{t}\left(x|z\right)}\left[s\right]=K\left(t\right)\hat{s}+\mu \left(t\right), \end{array}$$
where K(t) is defined by
(57)$$\begin{array}{c} K\left(t\right):=\left(\begin{array}{cc} O & \Sigma_{xz}\left(t\right)\Sigma_{zz}^{-1}\left(t\right)\\ O & I \end{array}\right). \end{array}$$

By applying Theorem 1 to the LQG problem with memory limitation, we obtain the following theorem:

**Theorem 3.** 
*In the LQG problem with memory limitation, the optimal control function of ML-POSC is provided by*

(58)
$$\begin{array}{c} u^{*}\left(t,z\right)=-R^{-1}B^{⊤}\left(\Pi K\hat{s}+\Psi \mu \right), \end{array}$$

*where K(t) (Equation 57) depends on Σ(t), and μ(t) and Σ(t) are the solutions of the following ordinary differential equations:*

(59)
$$\begin{array}{cc} \frac{d\mu }{dt} & =\left(A-BR^{-1}B^{⊤}\Psi \right)\mu , \end{array}$$


(60)
$$\begin{array}{cc} \frac{d\Sigma }{dt} & =\sigma \sigma^{⊤}+\left(A-BR^{-1}B^{⊤}\Pi K\right)\Sigma +\Sigma {\left(A-BR^{-1}B^{⊤}\Pi K\right)}^{⊤}, \end{array}$$

*where $\mu \left(0\right)=\mu_{0}$ and $\Sigma \left(0\right)=\Sigma_{0}$, while Ψ(t) and Π(t) are the solutions of the following ordinary differential equations:*

(61)
$$\begin{array}{cc} -\frac{d\Psi }{dt} & =Q+A^{⊤}\Psi +\Psi A-\Psi BR^{-1}B^{⊤}\Psi , \end{array}$$


(62)
$$\begin{array}{cc} -\frac{d\Pi }{dt} & =Q+A^{⊤}\Pi +\Pi A-\Pi BR^{-1}B^{⊤}\Pi +{\left(I-K\right)}^{⊤}\Pi BR^{-1}B^{⊤}\Pi \left(I-K\right), \end{array}$$

*where Ψ(T)=Π(T)=P.*


**Proof.** The proof is shown in Appendix E. □

Here, (Equation 61) is the Riccati equation [9,10,23], which appears in the LQG problem without memory limitation as well (Equation 37). In contrast, (Equation 62) is a new equation of the LQG problem with memory limitation, which in this paper we call the partially observable Riccati equation. Because estimation and control are not clearly separated in the LQG problem with memory limitation, the Riccati Equation (Equation 61) for control is modified to include estimation, which corresponds to the partially observable Riccati Equation (Equation 62). As a result, the partially observable Riccati Equation (Equation 62) is able to improve estimation as well as control.

In order to support this interpretation, we analyze the partially observable Riccati Equation (Equation 62) by comparing it with the Riccati Equation (Equation 61). Because only the last term of (Equation 62) is different from (Equation 61), we denote it as follows:(63)$$\begin{array}{c} Q:={\left(I-K\right)}^{⊤}\Pi BR^{-1}B^{⊤}\Pi \left(I-K\right). \end{array}$$

Q can be calculated as follows:(64)$$\begin{array}{c} Q=\left(\begin{array}{cc} P_{xx} & -P_{xx}\Sigma_{xz}\Sigma_{zz}^{-1}\\ -\Sigma_{zz}^{-1}\Sigma_{zx}P_{xx} & \Sigma_{zz}^{-1}\Sigma_{zx}P_{xx}\Sigma_{xz}\Sigma_{zz}^{-1} \end{array}\right), \end{array}$$
where $P_{xx}:={\left(\Pi BR^{-1}B^{⊤}\Pi \right)}_{xx}$. Because $P_{xx}⪰O$ and $\Sigma_{zz}^{-1}\Sigma_{zx}P_{xx}\Sigma_{xz}\Sigma_{zz}^{-1}⪰O$, $\Pi_{xx}$ and $\Pi_{zz}$ may be larger than $\Psi_{xx}$ and $\Psi_{zz}$, respectively. Because $\Pi_{xx}$ and $\Pi_{zz}$ are the negative feedback gains of the state *x* and the memory *z*, respectively, Q may decrease $\Sigma_{xx}$ and $\Sigma_{zz}$. Moreover, when $\Sigma_{xz}$ is positive/negative, $\Pi_{xz}$ may be smaller/larger than $\Psi_{xz}$, which may increase/decrease $\Sigma_{xz}$. A similar discussion is possible for $\Sigma_{zx}$, $\Pi_{zx}$, and $\Psi_{zx}$, as Σ, Π, and Ψ are symmetric matrices. As a result, Q may decrease the following conditional covariance matrix:(65)$$\begin{array}{c} \Sigma_{x|z}:=\Sigma_{xx}-\Sigma_{xz}\Sigma_{zz}^{-1}\Sigma_{zx}, \end{array}$$
which corresponds to the estimation error of the state from the memory. Therefore, the partially observable Riccati Equation (Equation 62) may improve estimation as well as control, which is different from the Riccati Equation (Equation 61).

Because the problem in Section 6.1 is specialized more than that in Section 6.2, we can carry out a more specific discussion. In the problem in Section 6.1, $\Psi_{xx}$ is the same as the solution of the Riccati equation of the conventional POSC (Equation 37), and $\Psi_{xz}=O$, $\Psi_{zx}=O$, and $\Psi_{zz}=O$ are satisfied. As a result, the memory control does not appear in the Riccati equation of ML-POSC (Equation 61). In contrast, because of the last term of the partially observable Riccati Equation (Equation 62), $\Pi_{xx}$ is not the solution of the Riccati Equation (Equation 37), and $\Pi_{xz}\neq O$, $\Pi_{zx}\neq O$, and $\Pi_{zz}\neq O$ are satisfied. As a result, the memory control appears in the partially observable Riccati Equation (Equation 62), which may improve the state estimation.

### 6.4. Comparison with Completely Observable Stochastic Control

In this subsection, we compare ML-POSC with COSC of the extended state. By applying Proposition 2 in the LQG problem, the optimal control function of COSC of the extended state can be obtained as follows:

**Proposition 4** ([10,23])**.**
*In the LQG problem, the optimal control function of COSC of the extended state is provided by*
(66)$$\begin{array}{c} u^{*}\left(t,s\right)=-R^{-1}B^{⊤}\Psi s=-R^{-1}B^{⊤}\left(\Psi \hat{s}+\Psi \mu \right), \end{array}$$
*where Ψ(t) is the solution of the Riccati Equation (Equation 61).*

**Proof.** The proof is shown in [10,23]. □

The optimal control function of COSC of the extended state (Equation 66) can be derived intuitively from that of ML-POSC (Equation 58). In ML-POSC, $K\hat{s}=E_{p_{t}\left(x|z\right)}\left[\hat{s}\right]$ is the estimator of the stochastic extended state. In COSC of the extended state, because the stochastic extended state is completely observable, its estimator is provided by $\hat{s}$, which corresponds to K=I. By changing the definition of *K* from (Equation 57) to K=I, the partially observable Riccati Equation (Equation 62) is reduced to the Riccati Equation (Equation 61), and the optimal control function of ML-POSC (Equation 58) is reduced to that of COSC (Equation 66). As a result, the optimal control function of ML-POSC (Equation 58) can be interpreted as the generalization of that of COSC (Equation 66).

While the second term is the same between (Equation 58) and (Equation 66), the first term is different. The second term is the control of the expected extended state μ, which does not depend on the realization. In contrast, the first term is the control of the stochastic extended state $\hat{s}$, which depends on the realization. The first term has two different points: (i) The estimators of the stochastic extended state in COSC and ML-POSC are provided by $\hat{s}$ and $K\hat{s}=E_{p_{t}\left(x|z\right)}\left[\hat{s}\right]$, respectively, which is reasonable because ML-POSC needs to estimate the state from the memory; and (ii) The control gains of the stochastic extended state in COSC and ML-POSC are provided by Ψ and Π, respectively. While Ψ improves only control, Π improves estimation as well as control.

### 6.5. Numerical Algorithm

In the LQG problem, the partial differential equations are reduced to the ordinary differential equations. The FP Equation (Equation 23) is reduced to (Equation 59) and (Equation 60), and the HJB Equation (Equation 28) is reduced to (Equation 61) and (Equation 62). As a result, the optimal control function (Equation 58) can be obtained more easily in the LQG problem.

The Riccati Equation (Equation 61) can be solved backwards from the terminal condition. In contrast, the partially observable Riccati Equation (Equation 62) cannot be solved in the same way as the Riccati Equation (Equation 61), as it depends on the forward equation of Σ (Equation 60) through *K* (Equation 57). Because the forward equation of Σ (Equation 60) depends on the backward equation of Π (Equation 62) as well, they must be solved simultaneously.

A similar problem appears in the mean-field game and control, and numerous numerical methods have been developed to deal with it [33]. In this paper, we solve the system of (Equation 60) and (Equation 62) using the forward–backward sweep method, which computes (Equation 60) and (Equation 62) alternately [33,34]. In ML-POSC, the convergence of the forward–backward sweep method is guaranteed [34].

## 7. Numerical Experiments

In this section, we demonstrate the effectiveness of ML-POSC using numerical experiments on the LQG problem with memory limitation as well as on the non-LQG problem.

### 7.1. LQG Problem with Memory Limitation

In this subsection, we show the significance of the partially observable Riccati Equation (Equation 62) by a numerical experiment of the LQG problem with memory limitation. We consider the state $x_{t}\in R$, the observation $y_{t}\in R$, and the memory $z_{t}\in R$, which evolve by the following SDEs: (67)$$\begin{array}{cc} dx_{t} & =\left(x_{t}+u_{t}\right)dt+d\omega_{t}, \end{array}$$(68)$$\begin{array}{cc} dy_{t} & =x_{t}dt+d\nu_{t}, \end{array}$$(69)$$\begin{array}{cc} dz_{t} & =v_{t}dt+dy_{t}, \end{array}$$
where $x_{0}$ and $z_{0}$ obey standard Gaussian distributions, $y_{0}$ is an arbitrary real number, $\omega_{t}\in R$ and $\nu_{t}\in R$ are independent standard Wiener processes, and $u_{t}=u\left(t,z_{t}\right)\in R$ and $v_{t}=v\left(t,z_{t}\right)\in R$ are the controls. The objective function to be minimized is provided as follows:(70)$$\begin{array}{c} J\left[u,v\right]:=E\left[\int_{0}^{10}\left(x_{t}^{2}+u_{t}^{2}+v_{t}^{2}\right)dt\right]. \end{array}$$

Therefore, the objective of this problem is to minimize the state variance by the small state and memory controls. Because this problem includes the memory control cost, it corresponds to the LQG problem with memory limitation.

Figure 2a–c shows the trajectories of Ψ and Π; $\Pi_{xx}$ and $\Pi_{zz}$ are larger than $\Psi_{xx}$ and $\Psi_{zz}$, respectively, and $\Pi_{xz}$ is smaller than $\Psi_{xz}$, which is consistent with our discussion in Section 6.3. Therefore, the partially observable Riccati equation may reduce the estimation error of the state from the memory. Moreover, while the memory control does not appear in the Riccati equation ($\Psi_{xz}=\Psi_{zz}=0$), it appears in the partially observable Riccati equation ($\Pi_{xz}\neq 0$, $\Pi_{zz}\neq 0$), which is consistent with our discussion in Section 6.3. As a result, the memory control plays an important role in estimating the state from the memory.

In order to clarify the significance of the partially observable Riccati Equation (Equation 62), we compare the performance of the optimal control function (Equation 58) with that of the following control function:(71)$$\begin{array}{c} u^{\Psi }\left(t,z\right)=-R^{-1}B^{⊤}\left(\Psi K\hat{s}+\Psi \mu \right), \end{array}$$
in which Π is replaced with Ψ. This result is shown in Figure 2d–f. In the control function (Equation 71), the distributions of the state and the memory are unstable, and the cumulative cost diverges. By contrast, in the optimal control function (Equation 58), the distributions of the state and memory are stable, and the cumulative cost is smaller. This result indicates that the partially observable Riccati Equation (Equation 62) plays an important role in the LQG problem with memory limitation.

### 7.2. Non-LQG Problem

In this subsection, we investigate the potential effectiveness of ML-POSC for a non-LQG problem by comparing it with the local LQG approximation of the conventional POSC [3,4]. We consider the state $x_{t}\in R$ and the observation $y_{t}\in R$, which evolve according to the following SDEs: (72)$$\begin{array}{cc} dx_{t} & =u_{t}dt+d\omega_{t}, \end{array}$$(73)$$\begin{array}{cc} dy_{t} & =x_{t}dt+d\nu_{t}, \end{array}$$
where $x_{0}$ obeys the Gaussian distribution $p_{0}\left(x_{0}\right)=N\left(x_{0}|0,0.01\right)$, $y_{0}$ is an arbitrary real number, $\omega_{t}\in R$ and $\nu_{t}\in R$ are independent standard Wiener processes, and $u_{t}=u\left(t,y_{0:t}\right)\in R$ is the control. The objective function to be minimized is provided as follows:(74)$$\begin{array}{c} J\left[u\right]:=E\left[\int_{0}^{1}\left(Q\left(t,x_{t}\right)+u_{t}^{2}\right)dt+10x_{1}^{2}\right], \end{array}$$
where
(75)$$\begin{array}{c} Q\left(t,x\right):=\left\{\begin{array}{cc} 1000 & (0.3\leq t\leq 0.6,0.1\leq |x|\leq 2.0),\\ 0 & (others). \end{array}\right. \end{array}$$

The cost function is high on the black rectangles in Figure 3a, which represent the obstacles. In addition, the terminal cost function is the lowest on the black cross in Figure 3a, which represents the desirable goal. Therefore, the system should avoid the obstacles and reach the goal with the small control. Because the cost function is non-quadratic, it is a non-LQG problem, which cannot be solved exactly by the conventional POSC.

In the local LQG approximation of the conventional POSC [3,4], the Zakai equation and the Bellman equation are locally approximated by the Kalman filter and the Riccati equation, respectively. Because the Bellman equation is reduced to the Riccati equation, the local LQG approximation can be solved numerically even in the non-LQG problem.

ML-POSC determines the control $u_{t}\in R$ based on the memory $z_{t}\in R$, i.e., $u_{t}=u\left(t,z_{t}\right)$. The memory dynamics is formulated with the following SDE:(76)$$\begin{array}{cc} dz_{t} & =dy_{t}, \end{array}$$
where $p_{0}\left(z_{0}\right)=N\left(z_{0}|0,0.01\right)$. For the sake of simplicity, the memory control is not considered.

Figure 3 is the numerical result comparing the local LQG approximation and ML-POSC. Because the local LQG approximation reduces the Bellman equation to the Riccati equation by ignoring non-LQG information, it cannot avoid the obstacles, which results in a higher objective function. In contrast, because ML-POSC reduces the Bellman equation to the HJB equation while maintaining non-LQG information, it can avoid the obstacles, which results in a lower objective function. Therefore, our numerical experiment shows that ML-POSC can be superior to local LQG approximation.

## 8. Discussion

In this work, we propose ML-POSC, which is an alternative theoretical framework to the conventional POSC. ML-POSC first formulates the finite-dimensional and stochastic memory dynamics explicitly, then optimizes the memory dynamics considering the memory cost. As a result, unlike the conventional POSC, ML-POSC can consider memory limitation as well as incomplete information. Furthermore, because the optimal control function of ML-POSC is obtained by solving the system of HJB-FP equations, ML-POSC can be solved in practice even in non-LQG problems. ML-POSC can generalize the LQG problem to include memory limitation. Because estimation and control are not clearly separated in the LQG problem with memory limitation, the Riccati equation can be modified to the partially observable Riccati equation, which improves estimation as well as control. Furthermore, ML-POSC can provide a better result than the local LQG approximation in a non-LQG problem, as ML-POSC reduces the Bellman equation while maintaining non-LQG information.

ML-POSC is effective for the state estimation problem as well, which is a part of the POSC problem. Although the state estimation problem can be solved in principle by the Zakai equation [38,39,40], it cannot be solved directly, as the Zakai equation is infinite-dimensional. In order to resolve this problem, a particle filter is often used to approximate the infinite-dimensional Zakai equation as a finite number of particles [38,39,40]. However, because the performance of the particle filter is guaranteed only in the limit of a large number of particles, a particle filter may not be practical in cases where the available memory size is severely limited. Furthermore, a particle filter cannot take the memory noise and cost into account. ML-POSC resolves these problems, as it can optimize the state estimation under memory limitation.

ML-POSC may be extended from a single-agent system to a multi-agent system. POSC of a multi-agent system is called decentralized stochastic control (DSC) [41,42,43], which consists of a system and multiple controllers. In DSC, each controller needs to estimate the controls of the other controllers as well as the state of the system, which is essentially different from the conventional POSC. Because the estimation among the controllers is generally intractable, the conventional POSC approach cannot be straightforwardly extended to DSC. In contrast, ML-POSC compresses the observation history into the finite-dimensional memory, which simplifies estimation among the controllers. Therefore, ML-POSC may provide an effective approach to DSC. Actually, the finite-state controller, the idea of which is similar with ML-POSC, plays a key role in extending POMDP from a single-agent system to a multi-agent system [22,44,45,46,47,48]. ML-POSC may be extended to a multi-agent system in a similar way as a finite-state controller.

ML-POSC can be naturally extended to the mean-field control setting [28,29,30] because ML-POSC is solved based on the mean-field control theory. Therefore, ML-POSC can be applied to an infinite number of homogeneous agents. Furthermore, ML-POSC can be extended to a risk-sensitive setting, as this is a special case of the mean-field control setting [28,29,30]. Therefore, ML-POSC can consider the variance of the cost as well as its expectation.

Nonetheless, more efficient algorithms are needed in order to solve ML-POSC with a high-dimensional state and memory. In the mean-field game and control, neural network-based algorithms have recently been proposed which can solve high-dimensional problems efficiently [49,50]. By extending these algorithms, it might be possible to solve high-dimensional ML-POSC efficiently. Furthermore, unlike the mean-field game and control, the coupling of HJB-FP equations is limited to the optimal control function in ML-POSC. By exploiting this property, more efficient algorithms for ML-POSC may be proposed [34].

## Figures and Tables

**Figure 1 entropy-24-01599-f001:**
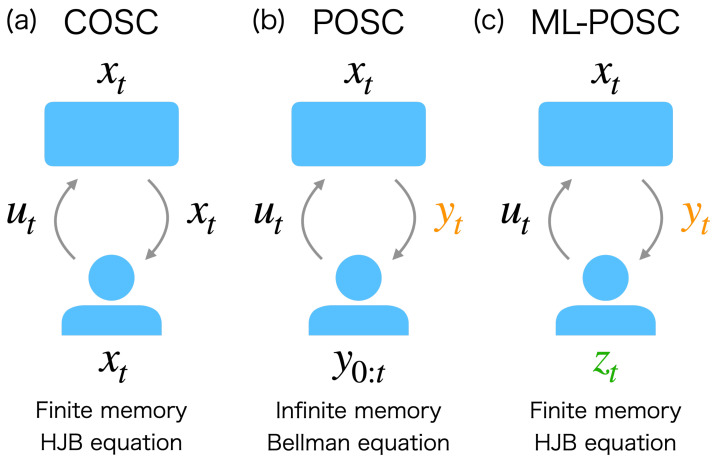
Schematic diagram of (**a**) completely observable stochastic control (COSC), (**b**) partially observable stochastic control (POSC), and (**c**) memory-limited partially observable stochastic control (ML-POSC). The top and bottom figures represent the system and controller, respectively; $x_{t}\in R^{d_{x}}$ is the state of the system; $y_{t}\in R^{d_{y}}$, $z_{t}\in R^{d_{z}}$, and $u_{t}\in R^{d_{u}}$ are the observation, memory, and control of the controller, respectively. (**a**) In COSC, the controller can completely observe the state $x_{t}$, and determines the control $u_{t}$ based on the state $x_{t}$, i.e., $u_{t}=u\left(t,x_{t}\right)$. Only finite-dimensional memory is required to store the state $x_{t}$, and the optimal control $u_{t}^{*}$ is obtained by solving the Hamilton–Jacobi–Bellman (HJB) equation, which is a partial differential equation. (**b**) In POSC, the controller cannot completely observe the state $x_{t}$; instead, it obtains the noisy observation $y_{t}$ of the state $x_{t}$. The control $u_{t}$ is determined based on the observation history $y_{0:t}:=\left\{y_{\tau }|\tau \in \left[0,t\right]\right\}$, i.e., $u_{t}=u\left(t,y_{0:t}\right)$. An infinite-dimensional memory is implicitly assumed to store the observation history $y_{0:t}$. Furthermore, to obtain the optimal control $u_{t}^{*}$, the Bellman equation (a functional differential equation) needs to be solved, which is generally intractable, even numerically. (**c**) In ML-POSC, the controller is only accessible to the noisy observation $y_{t}$ of the state $x_{t}$, as in POSC. In addition, it has only finite-dimensional memory $z_{t}$, which cannot completely memorize the the observation history $y_{0:t}$. The controller of ML-POSC compresses the observation history $y_{0:t}$ into the finite-dimensional memory $z_{t}$, then determines the control $u_{t}$ based on the memory $z_{t}$, i.e., $u_{t}=u\left(t,z_{t}\right)$. The optimal control $u_{t}^{*}$ is obtained by solving the HJB equation (a partial differential equation), as in COSC.

**Figure 2 entropy-24-01599-f002:**
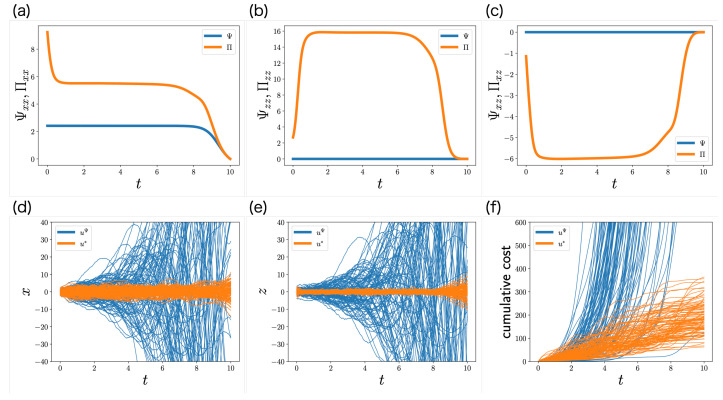
Numerical simulation of the LQG problem with memory limitation. (**a**–**c**) Trajectories of the elements of $\Psi \left(t\right)\in R^{2\times 2}$ and $\Pi \left(t\right)\in R^{2\times 2}$. Because $\Psi_{zx}\left(t\right)=\Psi_{xz}\left(t\right)$ and $\Pi_{zx}\left(t\right)=\Pi_{xz}\left(t\right)$, $\Psi_{zx}\left(t\right)$ and $\Pi_{zx}\left(t\right)$ are not visualized. (**d**–**f**) Stochastic behaviors of the state $x_{t}$ (**d**), the memory $z_{t}$ (**e**), and the cumulative cost (**f**) for 100 samples. The expectation of the cumulative cost at t=10 corresponds to the objective function (Equation 70). Blue and orange curves are controlled by (Equation 71) and (Equation 58), respectively.

**Figure 3 entropy-24-01599-f003:**
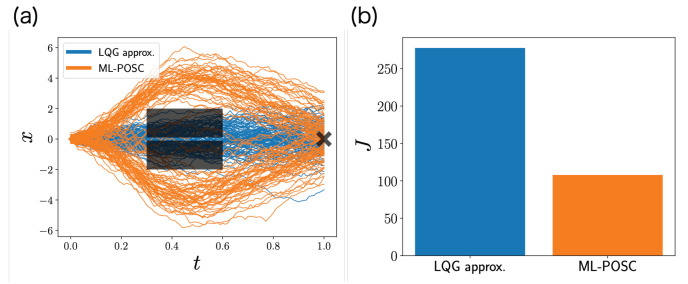
Numerical simulation of the non-LQG problem for the local LQG approximation (blue) and ML-POSC (orange). (**a**) Stochastic behaviors of state $x_{t}$ for 100 samples. The black rectangles and cross represent the obstacles and goal, respectively. (**b**) The objective function (Equation 74), computed from 100 samples.

## Data Availability

Not applicable.

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
