# Peer review of "Memory-Limited Partially Observable Stochastic Control and Its Mean-Field Control Approach"

_entropy, 2022, doi:10.3390/e24111599_

Round 1
Reviewer 1 Report
The paper “Memory-Limited Partially Observable Stochastic Control and its Mean-Field Control Approach”, by Takehiro Tottori and Tetsuya J. Kobayashi, introduces the construction of the memory-limited partially observable stochastic control (ML-POSC) for the Linear-Quadratic-Gaussian (LQG) problem, which can directly considers memory limitation as well as incomplete information. Such a control first formulates the finite-dimensional and stochastic memory dynamics explicitly, and then optimizes the memory dynamics considering the memory cost. It can be constructed by employing the mathematical technique of the mean-field control theory. Within this approach, the optimal control function of the ML-POSC is obtained by solving the system composed by the Hamilton-Jacobi-Bellman equation and the Fokker-Planck equation (the system of the HJB-FP equations). The forward-backward sweep method (fixed-point iteration method) is used to obtain the optimal control function of the ML-POSC, which computes the forward FP equation and the backward HJB equation alternately. In the mean-field game and control, the convergence of the forward-backward sweep method is not guaranteed, however, it is claimed that in the considered statement it is guaranteed in the ML-POSC because the coupling of the HJB-FP equations is limited to the optimal control function. Calculation results for two one-dimensional examples are provided via the ML-POSC: the first example deals with the LQG problem with memory limitation, and the second one discusses application to the non-LQG problem with phase constraints.
The paper contains new results on analysis of construction of the memory-limited partially observable stochastic control. These results can be useful to specialist in stochastic optimal control. One can recommend the paper for publication.
Remarks.
The numerical algorithm for the ML-POSC, particularly, the convergence of the forward-backward sweep method is guaranteed by results of the paper [34] (T. Tottori and T. J. Kobayashi in preparation, 2022), which according to References is not published yet.
Reviewer 2 Report
This paper proposed a memory-limited partially observable stochastic control (ML-POSC) framework which can consider memory limitation as well as incomplete information. Then, the Riccati equation is modified to the partially observable Riccati equation, which improves estimation as well as control.
The considered problem is interesting, and the motivation of this paper is clear. The following comments can be used to improve the current version.
In the Introduction part, the authors mentioned that POSC has three practical problems with respect to the implementation of the controller. However, proper literature is missing, which should be supplemented to support this point.
The major contributions of this paper should be further highlighted by comparing with some recently related works.
There exist some improper places in the use of the notations. For example, in (4), $T$ represents time horizon. However, in (94), $B^T$ represents the transpose of matrix $B$. Please unify the symbols appeared in this paper.
Is Theorem 1 a simple corollary of the results in [11,15]? If Yes, Theorem 1 should be reformulated as Lemma 1 or Corollary 1.
In Theorem 2, the Bellman equation (26) is a functional differential equation. It cannot be solved even numerically, which is the same problem as the conventional POSC. Therefore, the reviewer thinks Theorem 2 can be deleted in the revised version.
More remarks should be added to make the results of this paper more readable, such as Theorem 5 and Theorem 6.
The derivations of equation (81) and equation (89) should be further improved. Please add more details.
The format of the references cited in the article is not uniform, such as [34].
